# Characterization and Tribological Performances of Graphene and Fluorinated Graphene Particles in PAO

**DOI:** 10.3390/nano11082126

**Published:** 2021-08-20

**Authors:** Yanjie Chen, Enzhu Hu, Hua Zhong, Jianping Wang, Ayush Subedi, Kunhong Hu, Xianguo Hu

**Affiliations:** 1Department of Energy Materials and Chemical Engineering, Hefei University, Hefei 230601, China; chenyanjie0009@163.com (Y.C.); W964533917@163.com (J.W.); ayush.reticent19@gmail.com (A.S.); kunhonghu@163.com (K.H.); 2Department of Mechanical Engineering, Hefei University, Hefei 230601, China; hzhong@hfuu.edu.cn; 3Institute of Tribology, School of Mechanical Engineering, Hefei University of Technology, 193 Tunxi Road, Hefei 230009, China; xianguohu@hfut.edu.cn

**Keywords:** friction modifiers, solid lubricant additives, fluorocarbons, friction mechanisms

## Abstract

Graphene has been widely used as a lubricating additive to reduce the energy consumption of engines and improve fuel economy because of its unique crystal structure. Herein, graphene (GR) and fluorinated graphene (F-GR) nanoparticles were prepared by ball milling and liquid-phase exfoliation. The SEM/EDS, HRTEM, XPS, Raman spectrometer, X-ray spectrometer, FTIR were used to investigate the morphologies, surface groups, and crystal structure of two kinds of graphene materials. The influence of loads on the tribological properties of two kinds of particles was investigated in Poly Alpha Olefin (PAO6) using a UMT-2 reciprocating tribometer. Results showed that the crystal structure of GR is better than F-GR. F-GR can improve the lubrication performance of PAO6. For PAO6 containing 1 wt% F-GR at 10 N, the average friction coefficient and average wear rate decreased by 12.3% and 87% relative to pure PAO6, respectively. However, the high load resulted in an inconspicuous anti-wear and anti-friction effect. The influence of F-GR on the tribological behavior of PAO6 was more substantial than that of GR. The friction and wear mechanisms attributed to F-GR quickly entered the interface between the friction pairs. Friction-induced F-GR nanosheets mainly took the tribo-chemical reactions to participate in the lubrication film formation and helped achieve a low friction coefficient and wear rate.

## 1. Introduction

Lubricating oil additives have extreme anti-wear and anti-friction properties, which have been extensively explored by modern engine developers [1]. At present, traditional lubricating oil additives have several types, such as organic zinc dialkyl thiophosphate [2], molybdenum dialkyl thiophosphate [3], and solid lubrication additives of MoS_2_, WS_2_ and graphene, fullerene, and carbon nanotubes, which can modify the lubrication performance of lubricating oils [4,5,6,7].

Graphene (GR) materials have become a vital research spotlight. Since its discovery in 2004, GR has been widely studied due to its unique physical properties and broad application prospects as an essential additive for developing high-performance lubrication oils and greases [8]. Graphene has been widely used as a lubricating additive to reduce the energy consumption of engines and improve fuel economy [9,10]. Wang [11] added GR to SAE 10W-30 lubricating oil and studied the additive’s tribological properties as a lubrication modifier. The oil with 0.05 wt% graphene exhibited the minimum friction coefficient, lowest specific wear rate, and slightest scratch. Lin [1] utilized oleic acid and stearic acid to modify the surface of GR and hence achieved a good GR dispersion characteristic in 350SN base oil. The addition of modified GR at 0.075 wt% can substantially and steadily reduce the friction coefficient of 350SN base oil and enhance the oil’s anti-friction property. Studies indicated that modified GR quickly enters the friction contact surface to produce continuous friction adsorption film and prevents the direct contact of the friction surface; thus, a low friction coefficient and wear rate is achieved. Zhang [12] explored the tribological behavior of GR materials in oleic acid modified by a four-ball machine and found that adding a small amount of GR can improve the lubrication property of PAO. Zeng [13] studied the plasma interaction with the surface of SiO_2_ substrate to strengthen the adhesive attraction between GR and substrate and attain low friction for GR. Ramón-Raygoza [14] synthesized multilayer GR and loaded this material with copper particles. Results showed that graphite materials could improve the lubrication properties of base oil (SAE 25W-50). Zhao [15] noted that increasing the number of layers of GR worsens the tribological properties of GR materials. Friction induces a structural change, which shows the raised edge defects of GR. Liang [16] investigated the lubrication characteristics of in situ exfoliation graphite materials to enhance water-based lubricants. It discovered that improving the dispersion performance and friction interface can result in super-lubricity. To further improve the tribological properties of materials, scholars have increasingly explored GR material modification. Fluorinated GR (F-GR) is a new derivative of GR materials that was first reported in 2010. Increasing attention has been paid to F-GR materials due to the advantages of GR and F-GR [17,18]. Scholars realized the high efficiency of the lubricant additives because of their small size, high strength, and layered structure [19,20].

At present, some divergences in tribological mechanisms exist because only a few studies have investigated the tribology of F-GR. F-GR is the primary component unit of F-G, which combines some excellent properties of graphene and Fossilized fluorine ink, such as small size, high strength, and layered structure. It may become a more efficient lubricant [21]. Due to the introduction of F, the layer spacing of carbon atoms in molecular structure increases by nearly two times [22], resulting in the weakened interlayer forces. Moreover, the repulsion between fluorine atoms is also conducive to the sliding between layers, so its lubrication performance is better than that of graphene. At the same time, because FG has excellent chemical stability, high temperature, high pressure, and other harsh conditions can still maintain the ideal lubrication performance. Studies have found that FG applied in lubrication as an additive of lubricating oils will significantly reduce the friction coefficient of lubricating oils [23]. Hou [24] produced F-GR from fluorinated graphite and studied the effects of this material on the tribological performances of polyolefins (PAO40). The author found that the friction coefficient was not changed considerably, but the anti-wear property was improved remarkably. The best addition amount of F-GR was 0.25 mg/mL. F-GR composite coatings with different fluorine contents have been prepared, and the influence of coating structure and fluorine content on the lubrication performance of F-GR has been investigated [14]. Sun [21] discovered that F-GR materials could not obviously improve composite’s mechanical and tribological properties but positively affect the mold resistance of materials. Li [25] examined the tribological properties of GR and F-GR and observed that the tribological properties of F-GR could be five to nine times better than those of GR when the fluorine content is adjusted. The cause of friction enhancement is the fold of high interfacial charge derived from the concentration of fluorine atoms in the area. Their results differ considerably from our own.

Given the literature above, certain differences in tribological properties exist between GR and F-GR. Thus, the study of friction mechanism remains to be developed. Herein, GR and F-GR materials were prepared using established methods to explore their lubrication addition to poly α-olefin base oil (PAO6), compare the additives’ tribological behaviors, and intensively analyze their friction mechanisms [26]. The effective implementation of this work not only provides theoretical guidance but can also benefit the development of high-performance lubricants.

## 2. Experimental

### 2.1. Raw Materials and Instruments

The commercial graphite powder (Aladdin Industrial, Shanghai, China) (micron-grade grain size, 3–5 μm), fluorinated graphite (XFNANO, Inc., Nanjing, China, advanced materials supplier) (thickness less than 10 nm), ethyl alcohol (C_2_H_5_OH), isopropanol ((CH_3_)_2_CHOH) (Tianjin Zhiyuan Reagent Co., Ltd., Tianjin, China), PAO6, N-methyl pyrrolidone (NMP Wuxi Yatai United Chemical Co., Ltd., Wuxi, China) were all of the analysis reagents. GR and F-GR were prepared as previously described [26,27]. The detailed processes are as follows. A total of 500 mg graphite powder was added to the surfactant NMP. Then, in a 100 mL beaker, we added 50 mL isopropyl alcohol to the mixture as the dispersant and stirred with a glass rod until evenly dispersed. The mixture was transferred into a planetary ball (Model PM-0.4) milling machine with 10 mm-diameter steel balls for 5 h of milling. Afterward, the sample was drawn out with a dispette and ultrasonicated for 2 h. The sample was centrifuged using a high-speed centrifuge machine (Model HC-2064) for 30 min. After the supernatant solution was removed, the precipitate in the centrifugal tube was dried in a drying cabinet (Model TY-ZK-1) for 2 h to eliminate the residual NMP. Dried GR powder was obtained.

F-GR was prepared by weighing 500 mg fluorine fossil ink, 200 mg PVP (Shanghai Zhanyun Chemical Co.,Ltd., Shanghai, China) as a surfactant put into the ball grinding tank, according to the ball material ratio with a diameter of 13 mm, 6 mm, and 5 mm ratio of 8:15:17. Add 50 mL isopropanol as dispersant and grind it in a planetary mill at 400 RPM for 10 h. After that, take out the product and ultrasonic it in an ultrasonic cleaning machine for 2 h. After the ultrasound, the product was centrifuged at 2000 r/min for 30 min. After centrifugation, the residue was extracted, and the precipitate in the centrifuge tube was dried in a drying oven (2 h model Ty-ZK-1) to eliminate the remaining PVP. Fossilized fluorine alkene powder was obtained.

### 2.2. Analysis Method

The effects of different loads (5, 10, 15, 50 and 150 N) on the lubrication properties of PAO6 added with varying contents of GR and F-GR were investigated using a UMT-2 reciprocation tribometer [28]. The remaining test conditions included the reciprocation speed of 10 mm/s and a stroke length of 5 mm and a test time of 60 min at room temperature. The Hertz contact stresses were calculated via single point contact model [29,30]. They are 0.379 GPa (5 N), 0.707 GPa (10 N), 1.06 GPa (15 N), 1.21 GPa (50 N), 0.808 GPa (150 N), respectively.

A commercially available bearing steel ball (10 grade, Chinese standard) with a diameter of 6 mm was used as the stationary upper counterpart. The hardness of the steel ball was 60–63 RHC. The lower specimens were GCr15 steel disks with a diameter of 28 mm, a thickness of 2 mm, and a surface roughness of 0.03 μm. The hardness of the steel disk was 50–60 RHC. All friction pairs were ultrasonically cleaned with ethanol for 5 min before the friction tests. A detailed schematic of the tribometer is shown in Figure 1.

All friction tests were carried out three times under the same conditions to ensure repeatability and reliability. The average friction coefficient and average wear rate were calculated based on the repeated tests; besides, the relative deviation of tests should be lower than 5%. However, some tests for calculating the average wear rates may be higher than 5% due to the tester’s accuracy, which could also be used to reflect the efficacy of modified lubrication property of PAO6, including the proper contents of GR and F-GR.

The 3D laser scanning microscopy (Model VK-X100K) was used to investigate the wear profile of the steel ball and disks. The multifile analyzer was used to calculate the wear scar diameter, wear width, and wear area. The wear rate of the disks was evaluated using the following formula [31].
*Ko* = (*S_a_* × *A*)/*L* × *N*(1)

*Ko*: wear rate, (mm^3^/Nm); *S_a_*: wear area, μm^2^; *A*: amplitude, mm; *L*: sliding distance, m; *N*: load, N.

Four types of carbon materials were analyzed by X-ray diffraction (XRD; XRD-7000S/L). The specific surface area of GR and F-GR were tested by a basic dynamic chemisorption analyzer (ChemiSorb 2720). The specific surface areas of GR and F-GR were 11.0134 m^2^/g and 226.67 m^2^/g. The result indicated that the F-GR was easier to be adsorbed on the surface of friction pairs compared to that of GR. The micro-morphologies of GR and F-GR were analyzed by scanning electron microscopy (SEM; model JSM-6700F) and high-resolution transmission electron microscopy (HRTEM; JEOL model 2010). Fourier transform infrared (FTIR) absorption spectroscopy (Nicolet 6700 model) was used to determine the types of surface groups of the four-carbon materials. The element type, content, and distribution of the four-carbon materials and wear traces of the disks were characterized by an energy spectrum analyzer (energy-dispersive spectroscopy (EDS), model JSM-6700F). Raman spectroscopy (LabRAM-HR, resolution = 0.6 cm^−1^; scanning repeatability = ±0.2 cm^−1^) and XPS were utilized to investigate the element chemical valence state of disks lubricated with PAO6. Different contents of GR and F-GR were analyzed, and the friction and wear mechanisms of the two GR materials were further clarified [32,33,34,35].

## 3. Results and Discussion

### 3.1. Characterization of Carbon Materials

Figure 2 shows the SEM and HRTEM images of the four-carbon materials. Figure 2a reveals that the raw graphite powder possessed a flake structure with an un-uniform size with the crude surface and numerous obvious folds. Compared with graphite, the prepared GR exhibited a flake structure, and its size and thickness were smaller than those of the raw graphite. The increased transparency and surface smoothness, and decreased thickness of GR indicate that the ball-milling process successfully exfoliated the graphite (Figure 2b). HRTEM image (Figure 2c) also verified the successful preparation of GR due to the obvious GR flake structure. The selected diffraction pattern shows that GR was successfully prepared and achieved an obvious crystal structure, including the 002 and 004 crystal faces. The layer space of standard graphite was 0.34 nm [32]. The lattice spacing of four-layered GR was 1.336.

The surface of raw fluorinated graphite was rough and showed folds (Figure 2e). The surface of the prepared F-GR was smoother, more transparent, and thinner than that of fluorinated graphite. The integrity of F-GR was damaged to some extent; the size was reduced, and some fragments were deposited on large-scale laminates (Figure 2f). Results show that F-GR was successfully prepared by liquid-phase exfoliation. HRTEM images and selected diffraction patterns (Figure 2g,h), Large sheets of fluorine fossil ink can be seen in the HRTEM images, indicating that the separation of fluorine fossil ink layers is realized through liquid phase stripping, and the obtained FG sheets have a large transverse size. In the 100 nm HRTEM image, it is found that the monolayer FG is a relatively flat plane with high transparency, indicating that the material of FG film is relatively uniform and has good integrity. It can also be observed from the figure that the color was slightly darker locally, which was the overlap between the fluorinated graphite sheets due to the accumulation. These analyses could verify the successful preparation of F-GR.

Figure 3a displays the XRD spectra of the four-carbon samples. Graphite powder had four obvious diffraction peaks. The peaks at 12.5°, 26.6°, 44°, and 54° were attributed to the graphite (001), (002), (101), and (004) crystal plane diffraction peaks, respectively [17,36]. The spectrum of the prepared GR materials reveals that the material had 12° (001) and 44° (101) peaks, which were the characteristic peaks of GR. The intensities of the diffraction peaks of GR were weaker and broader than those of graphite; this discrepancy indicates that the particle size shrank and the integrity of the crystal structure decreased. These results showed the successful preparation of GR materials.

Fluorinated graphite had three obvious diffraction peaks. The diffraction peaks at 26.5°, 44°, and 54° were attributed to 002, 101, and 004 crystal faces [18]. The prepared F-GR possessed two characteristic peaks located at 12.5° and 44° [19]. These peaks were broad and weak, which indicated that the F-GR was successfully prepared.

Figure 3b shows the FTIR analysis of the four-carbon materials. In general, the peak at 1587 cm^−1^ belonged to the characteristic peak of benzene ring vibration. The peak at 1355 cm^−1^ was attributed to the bending vibration of −CH_3_. The peak at 3400 cm^−1^ was ascribed to the characteristic peak of –OH [24]. The peaks at 2951 and 2850 cm^−1^ were attributed to the stretching vibrations of –CH_3_ and –CH_2_–, respectively. The peak at 1218 cm^−1^ was ascribed to the –C–F group.

The intensities of the two peaks (2951 and 2850 cm^–1^) of graphite and fluorinated graphite were lower than those of GR and F-GR. These findings imply that the edge defects of GR and F-GR increased during the preparation process; this rise in edge defect can be ascribed to the graft of the liquid molecule [13]. The positions of the remaining peaks were the same as those of graphite powder; this result indicated that no obvious chemical reaction occurred during the ball grinding process. The graphite was physically and mechanically stripped to form GR materials.

The absorption peaks at 1215 and 1340 cm^−1^ respectively correspond to the characteristic absorption of covalent –CF– bonds [32]. Similarly, F-GR prepared by liquid-phase exfoliation had no other substantial changes in the peak positions of the two spectra, and this finding suggested that no remarkable chemical change occurred in the stripping process.

Figure 4 is the Raman analysis diagram of the four-carbon samples. As can be seen from Figure 4a, there was a prominent G peak in FG. In FGR, the characteristic absorption peak of acyl azide may be near the 2781 cm^−1^ peak. Firm absorption peaks appeared at 1404 cm^−1^ and 1510 cm^−1^, which were respectively assigned to the D peak and G peak. The relative intensity ratio of the D peak and G peak (I_D_/I_G_) was 0.9298. In Raman spectral analysis, I_D_/I_G_ could be used to characterize the disordered degree of the structure of graphene and I_D_/I_G_ was larger. It indicated that the degree of structural disorder was higher [33,34]. The FGR disordered structure can be clearly seen. As can be seen in Figure 4b, graphene exhibited obvious 2D peaks and G peaks in GR. The 2D peak of the graphene film was near 2700 cm^−1^. The 2D peak is derived from the dual resonant electron-photon scattering process, and its peak position and intensity were used to identify the number of graphene layers. The G peak near 1580 cm^−1^ was the characteristic peak of sp2 hybrid structure carbon, reflecting the symmetry and crystallization degree of graphene materials, indicating that its GR lamellar structure was excellent. In graphite, the slightly weaker D-peak was located near 1350 cm^−1^, whereas there was no D-peak in graphene because it is not present in the non-defective graphene [35].

### 3.2. Anti-Wear and Anti-Friction Performances

Graphene has excellent mechanical properties because the shear force between the sheet and the spread is minimal. It has a lower friction coefficient than graphite in theory, so it has also received significant attention in anti-friction lubrication. Figure 5 shows variations in the average friction coefficient and average wear rate of disks lubricated with PAO6, including the different contents of GR and F-GR at different loads. The average friction coefficient of pure PAO6 was 0.102 under the load of 5 N. With the addition of 0.1, 0.5, and 1 wt% of GR, the average friction coefficients of the oil sample all increased. For the F-GR, the average friction coefficients of the oil samples all decreased. Among these contents, 1 wt% F-GR had the more significant change rates of 16.9%.

At 10 N, the average friction coefficient of pure PAO6 was 0.10. The average friction coefficient also increased for GR. However, the average friction coefficients were all decreased for F-GR and ranged from 3% to 12%. Meanwhile, the oil samples with 1 wt% F-GR showed the most significant decrease in friction coefficient (12.3%). When the load increased to 15 N and 50 N, the average friction coefficient of the oil sample, including 1 wt% F-GR, changed much more than those of PAO6 containing different contents of GR and F-GR. When the loads were increased up to 150 N, the change rate of the average friction coefficient of the oil samples, including the F-GR, became small. These results indicated F-GR could be used as an additive in PAO.

Figure 5b reveals the average wear rates of the disks lubricated with different oil samples. The average wear rate of the disk lubricated with pure PAO6 at the load of 5 N was 0.26 × 10^−9^ mm^3^/Nm. The wear rate was substantially increased when different amounts of GR were added to the PAO6. However, the wear rates were all reduced for F-GR in PAO. F-GR at 1 wt% addition had the highest reduction in wear rate (0.033 × 10^−9^ mm^3^/Nm) with drops of 87%.

At 10 N, all graphene containing oils exhibited lower wear than PAO only. The average wear rates of the disks lubricated with PAO6 combined with 0.1 wt% GR and F-GR were 1.03 × 10^−9^ and 0.19 × 10^−9^ mm^3^/Nm, respectively, to 15.1% and 84.2% reductions relative to those of pure PAO6. When the load continued to increase to 15, 50, and 150 N, some of the wear rates of the disks lubricated with PAO6 combined with GR increased. This is mainly because the excess graphene flakes could agglomerate with the metal wear debris, making the contact surface rough and reducing wear resistance.

As shown in Figure 5c, a pure PAO6 oil sample contained graphene. The friction coefficient of the 0.1 wt% F-GR oil sample increased rapidly due to the oil film cracking in the friction process, which weakened the lubrication effect of base oil. The surface roughness was different, after stable friction coefficient, and when the F-GR amount of adding 0.5 wt% and 1 wt%, the friction coefficient of flat all the way, this may be due to the thickness of graphite fluoride nanoscale making it easier to enter the protective film formed between the friction pair. Lubricant oil film formed by joint action slows down the oil film rupture. With the increase of friction time, it was found that the friction coefficient of 0.5 wt% F-GR oil sample was less than 1 wt% F-GR oil sample, which may be because the amount of working fluorine fossil ink or fluorine fossil ink is inevitable under the set test conditions. The lubricating film formed by the lubricating oil with this additional amount was sufficient to withstand the shear force under the test condition. Excessive addition did not further enhance the tribological properties of the lubricating oil. The lubrication performance of F-GR modified PAO6 was better than that of GR. F-GR improved the wear resistance of PAO6, and F-GR could be used as a low-load lubricant additive of PAO6.

Figure 6 shows the optical microscope images of the wear traces of steel balls and disks lubricated with different oil samples at 50 N and 10 mm/s for 60 min. The addition of F-GR effectively reduced the wear scar diameter of the steel ball. The average wear scar diameter (AWSD) of the steel ball lubricated with pure PAO6 at 50 N was 263.5 μm. When 0.1 wt% F-GR was added, the AWSD decreased to 195.5 μm, which corresponded to a 25.8% reduction with respect to pure PAO6. The AWSD was 200.5 μm (reduced by 23.9%) when 0.5 wt% F-GR was added to PAO6. When 1 wt% F-GR was added to PAO6, the AWSD decreased to 187.5 μm (decreased by 28.8%). When different contents of GR particles (0.5 and 1 wt%) were added to PAO6, the AWSDs decreased to 247.3 μm (by 6.1%) and 221.3 μm (by 16%), respectively.

Figure 7 shows the wear morphology of the disc studied by a 3D laser scanning microscope (Model VK-x100K). Use multiple files to analyze wear area. At 50 N, the wear zone of pure PAO6 lubricated steel plate with different Gr and F-Gr contents is 136.7 μm^2^. The wear area was 162.1 μm^2^ (increased by −18.5%) when 0.1 wt% GR was added to PAO6 relative to pure PAO6. When 0.5 and 1 wt% GR were added, the wear areas were 104.5 and 178.3 μm^2^, respectively. When 0.1 wt% F-GR was added, the wear area was 68.6 μm^2^ and decreased by 49.8% relative to pure PAO6. When 0.5 and 1 wt% were added, the wear areas were 70.6 and 28.3 μm^2^, respectively. It shows that the fluoridation of graphene can improve the mechanical properties, physical properties, and optical properties of carbon nano-materials and improve their wear resistance and friction resistance.

### 3.3. SEM/EDS Analysis

Figure 8 shows the SEM/EDS analysis of the wear traces of disks lubricated with different oil samples at a load of 50 N. The deepest furrows appeared on the surface of steel disks lubricated with pure PAO6 (Figure 8a). When GR and F-GR were used as additives, both diminished the wear depth and furrow width (Figure 8b,c). The surface modification degree of disks lubricated with PAO containing the different F-GR was better than GR’s. The minor wear was attained under 1 wt% F-GR addition, the wear surface of the steel sheet was smoothened, and the number of parallel furrows was substantially reduced. The fluorine element was checked on the worn zones of disks lubricated with the different contents of F-GR in PAO, which indicated the tribofilm was formed on the surface to the resistance of friction and wear. Therefore, F-GR can enhance the anti-wear and anti-friction performances of base oil when used as lubrication additives.

Table 1 studies the atomic element content of disc wear traces of different oil samples when the load is 50 N, 25 mm/s, and the lubrication time is 30 min. The decrease of Fe content indicated that the addition of 0.1 wt% F-GR could effectively form a protective film on the steel plate’s surface to achieve wear resistance and friction reduction

### 3.4. Friction and Wear Mechanism Analysis

Microstructure analysis of the GR and F-GR nano-sheets showed their lamellar structures (Figure 2). The two types of GR materials could be adsorbed preferentially on friction pairs at low load conditions. This effect resulted in a sizeable interfacial space between the friction pairs during friction [36]. The sliding role of the nano-sheets on the interface of the friction pairs in this study resulted in a low friction coefficient and wear rate. The other reason is the formation of tribofilm on the surface of friction pairs. However, the average friction coefficient and wear rate of PAO6, including the different GR content, was increased due to the accumulation of GR. For F-GR, Friction induced the formation of tribofilm on the surface of worn zones resulted in the lower average friction coefficients and wear rates (Figure 3). Besides, the higher loads resulted in the lower wear rates for GR and F-GR as shown in Figure 5b. The potential mechanism related to the hardness, and Young modulus of nanomaterials [37,38]. In general, the high loads caused the accumulation of friction heat on the surface of friction-pairs which possibly promoted the occurrence of tribological chemical reactions which resulted in lower wear rates for samples at high loads. Moreover, the specific surface area analysis shows the F-GR was easier to be adsorbed on the surfaces of friction pairs which resulted in the low friction coefficient and wear rates.

As shown in Figure 9, the Raman method studied the wear marks containing 0.5 wt% GR and F-GR. It was found that after friction with GR and FGR in PAO6, G and D peaks, 2D peaks and iron oxides appeared in the wear marks, which indicated that in the process of friction and wear, wear debris was accumulated. FGR and GR lubricating oil films were generated.

XPS was performed to investigate the element chemical valence of the wear traces lubricated with PAO6 containing 0.1 wt% of GR and F-GR. Figure 10 shows the C1s spectrum of wear traces lubricated with different oil samples at 15 N. The peak at 284.2 eV was attributed to Csp2 and 288.5 eV to Csp3. The peak at 288.7 eV belonged to –COOH. The Csp2 (70.8%) content of the wear trace lubricated with pure PAO6 was lower than those of PAO6 containing GR (83.3%) and F-GR (78.4%). However, the Csp3 content was increased. The –COOH content was higher than those of PAO6 containing GR and F-GR. Results reflected that the friction-induced the GR and F-GR to participate in the formation of lubrication film. The carbon content in the tribofilm increased when GR and F-GR were added into PAO6, which can be used to clarify the results (Table 1). Moreover, the O1s spectrum shows the oxygen element chemical valence of wear traces of disks lubricated with different oil samples (Figure 10d–f). In general, the peaks at 529.2, 539, and 531.3 eV belonged to Fe_2_O_3_, F_3_O_4_, and FeOOH [39]. The FOOH was readily observed to increase when GR and F-GR were added to PAO6. These results indicated that the GR and F-GR in PAO6 caused the variation in the chemical composition of the lubrication film. Figure 7g reveals the Fe2p spectrum of the wear traces for different oil samples. The Fe1/2 peaks at 709–711 eV were attributed to the iron oxides, such as FeO, Fe_2_O_3_, and Fe_3_O_4_. The Fe1/2 peak at 707 eV was attributed to the Fe atom [29]. In Figure 10g, a small peak was detected at 707 eV on the disk lubricated with PAO6 + 0.1 wt% F-GR. This result indicated that the lubrication film protected the iron-based materials.

Figure 10h shows the N1s spectrum of wear traces lubricated with different oils. The nitrogen peak was not detected on the surface of the disk lubricated with pure PAO6. However, this peak was detected on the surface of disks lubricated with PAO6 + 0.1 wt% GR and PAO6 + 0.1 wt% F-GR. The N element was derived from GR or F-GR particles and resulted in the preparation of particles. Figure 10i shows the F1s spectrum of wear traces lubricated with different oils. The fluorine element was checked on the worn zone of disks lubricated with PAO containing 0.1 wt% contents of F-GR, which could be used to clarify the reduction of average friction coefficient and wear rate. These results revealed that the F-GR participated in the formation of lubrication film and led to a lowered friction coefficient and wear rate [40,41,42].

## 4. Conclusions

GR and F-GR were successfully prepared using graphite and fluorinated graphite as raw materials in order to develop an essential additive for developing high-performance lubrication oils and greases. The effects of GR and F-GR on the lubrication characteristics of PAO6 were studied using a UMT-2 Tribometer. The SEM/EDS, HRTEM, XPS, Raman spectrometer, X-ray spectrometer, and FTIR were used to analyze the morphology and structure of GR and F-GR. The conclusions achieved are as follows:

(1)GR has a better crystal structure than that of F-GR.(2)Two kinds of materials can improve the lubrication characteristics of PAO6 to some extent. The optimum addition level of F-GR was 1 wt% at 10 N, which resulted in the reduction of friction coefficient of 0.088 and wear rates of 0.033 × 10^−9^ mm^3^/Nm, which corresponded to decreases of 12.3% and 87%, respectively, relative to that of pure PAO6.(3)The wear resistance and friction reduction properties of F-GR for PAO6 were superior to GR. The accumulation of GR in PAO6 increased the average friction coefficient and wear rates at some conditions. Addition of graphite fluoride chips because of their small size, and a reasonable shear force quickly enter the wear interface achieving the effect of anti-wear and form a protective layer. The friction produced in the process of graphite fluoride debris can also be filled to contact surface grinding marks and bear part of the load to reduce wear and improve the effectiveness of the bearing capacity.

In all, this study will give a base referred value for application of GR and F-GR in extreme lubrication conditions such as high temperatures and loads.

## Figures and Tables

**Figure 1 nanomaterials-11-02126-f001:**
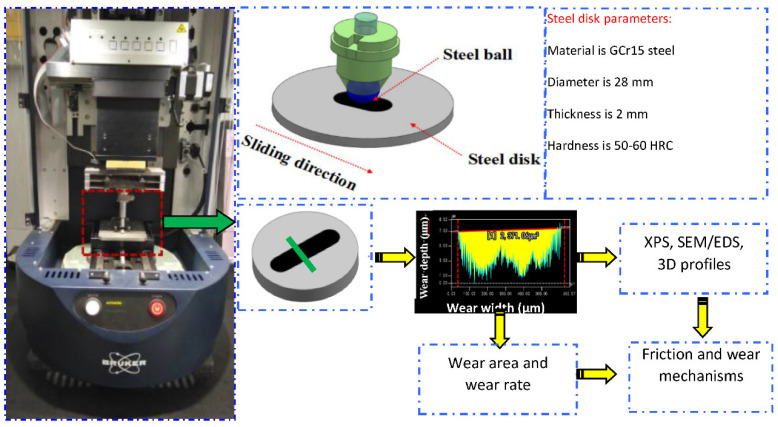
Schematic of the tribometer and analysis methods.

**Figure 2 nanomaterials-11-02126-f002:**
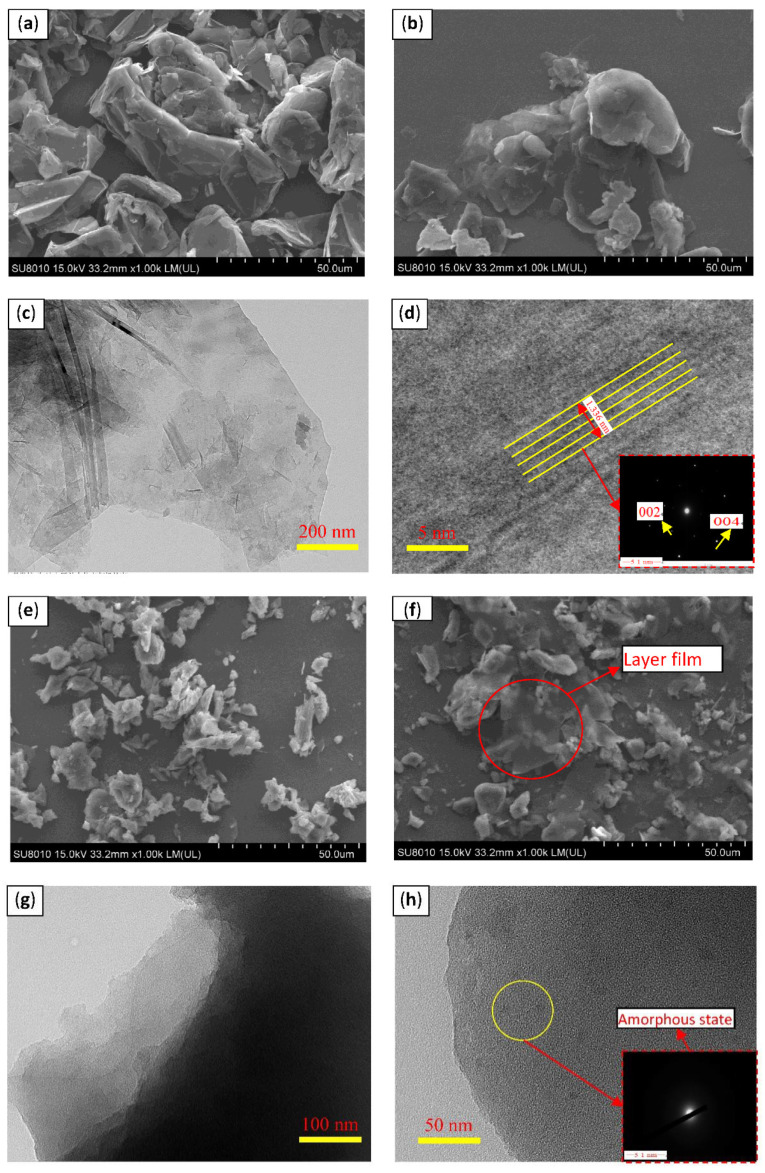
SEM and HRTEM images of different carbon materials. (**a**) SEM image of graphite, (**b**) SEM image of graphene, (**c**,**d**) HRTEM image of graphene and selected diffraction pattern, (**e**) SEM image of fluorinated graphite, (**f**) SEM image of fluorinated graphene (**g**,**h**) HRTEM image of fluorinated graphene and selected diffraction pattern.

**Figure 3 nanomaterials-11-02126-f003:**
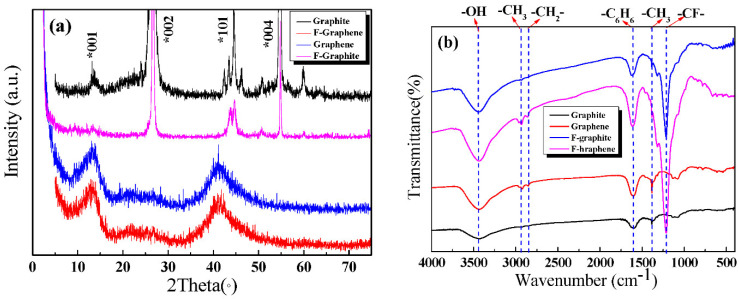
The XRD and FTIR analysis of four carbon materials. (**a**)XRD, (**b**) FTIR.

**Figure 4 nanomaterials-11-02126-f004:**
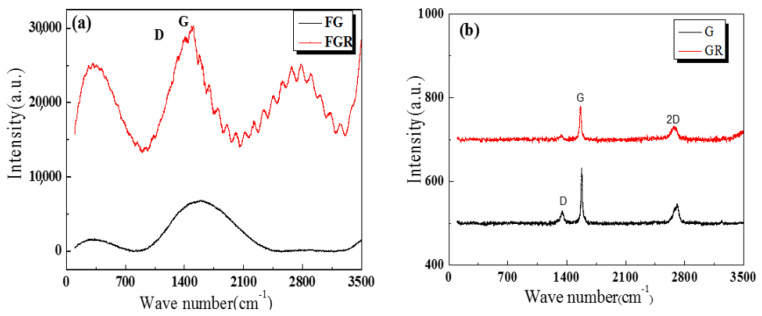
Raman analysis of four carbon materials. (**a**) Raman analysis of FG and FGR, (**b**) Raman analysis of G and GR.

**Figure 5 nanomaterials-11-02126-f005:**
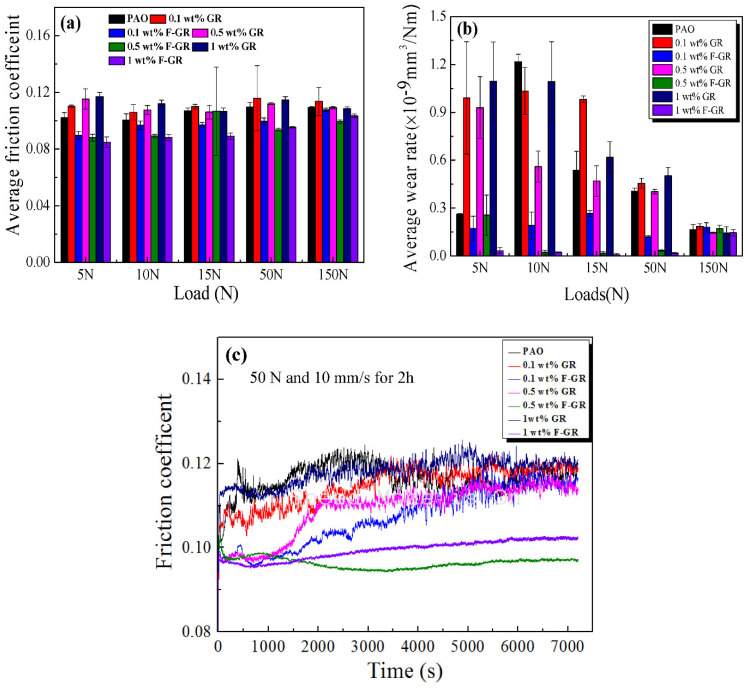
Variation of average friction coefficient and wear rate for different oil samples lubricated at 50 N and 10 mm/s for 60 min. (**a**) Average friction coefficient, (**b**) Average wear rate, (**c**) Friction coefficient curve.

**Figure 6 nanomaterials-11-02126-f006:**
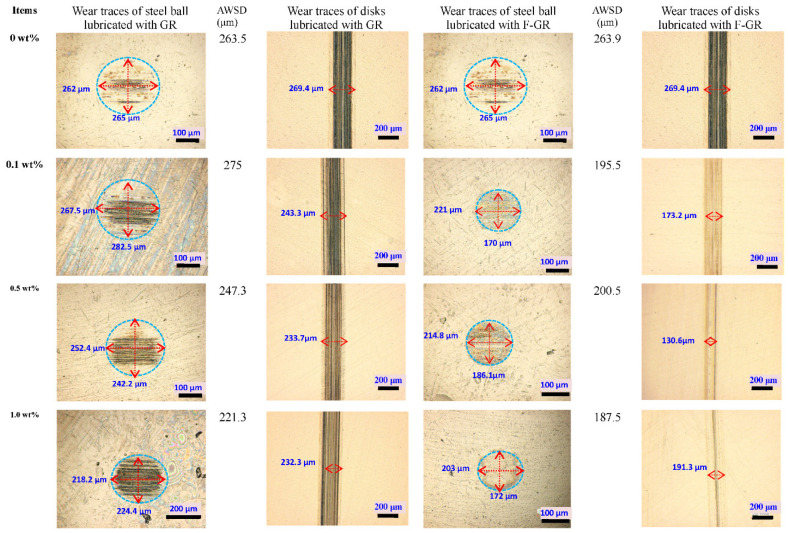
Optical morphologies of wear traces on the surfaces of steel balls and disks lubricated with different oil samples at 50 N and 10 mm/s for 60 min.

**Figure 7 nanomaterials-11-02126-f007:**
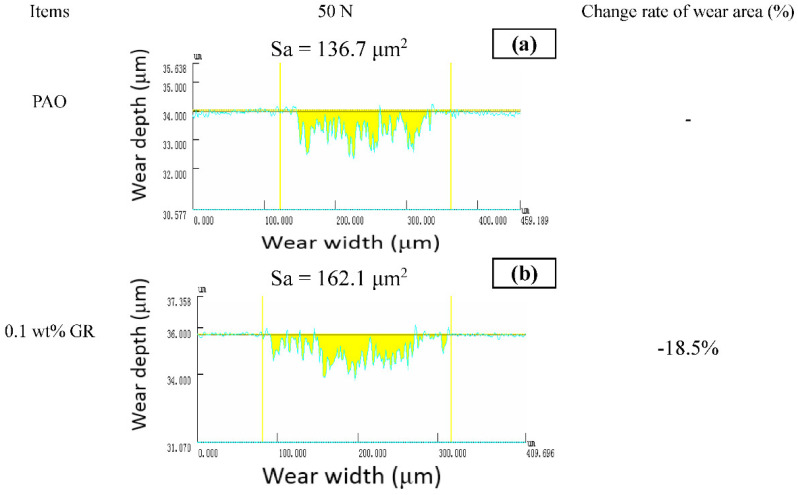
Crossing areas of wear traces of disks lubricated with different oil samples at 50 N and 10 mm/s for 60 min. (**a**) PAO6, (**b**) 0.1 wt% GR, (**c**) 0.1 wt% F-GR, (**d**) 0.5 wt% GR, (**e**) 0.5 wt% F-GR, (**f**) 1 wt% GR, (**g**) 1 wt% F-GR.

**Figure 8 nanomaterials-11-02126-f008:**
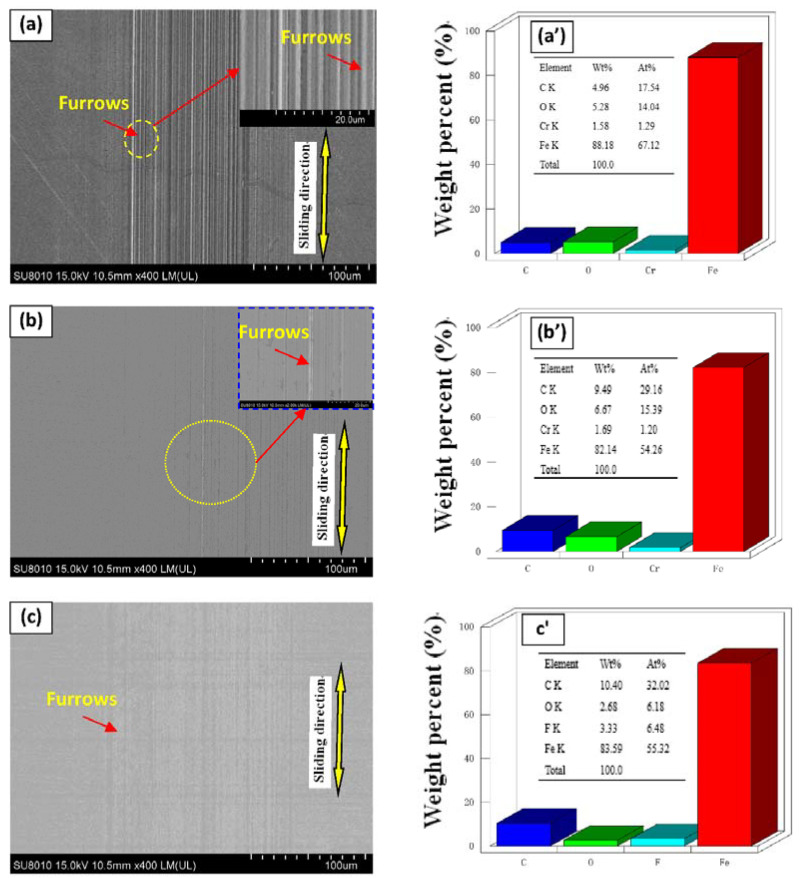
SEM/EDS analysis of steel sheet surface with different content of GR and F-GR added in PAO at 50 N and 10 mm/s for 60 min. (**a**) PAO6, (**b**) 0.1 wt% GR, (**c**) 0.1 wt% F-GR, (**d**) 0.5 wt% GR, (**e**) 0.5 wt% F-GR, (**f**) 1 wt% GR, (**g**) 1 wt% F-GR.

**Figure 9 nanomaterials-11-02126-f009:**
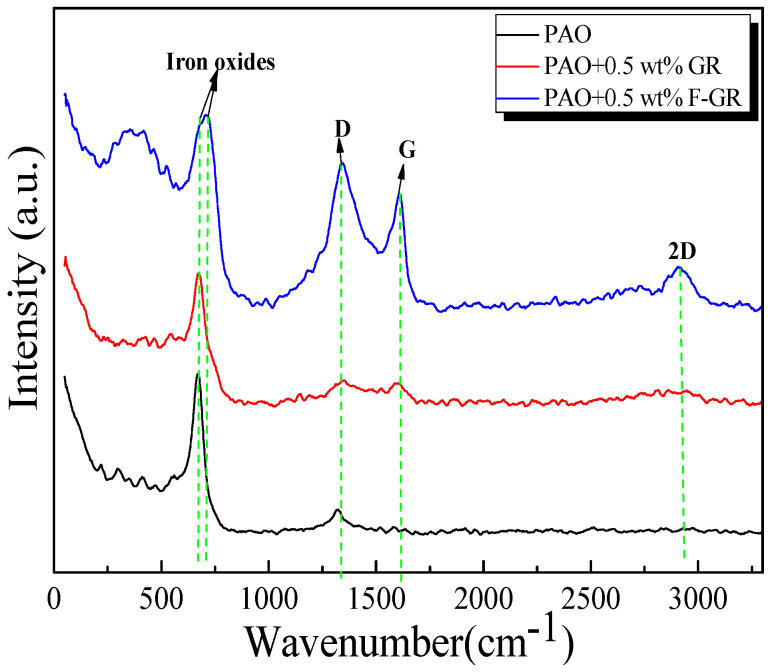
Raman analysis of worn traces of disks lubricated with PAO including 0.5 wt% GR and 0.5 wt% F-GR.

**Figure 10 nanomaterials-11-02126-f010:**
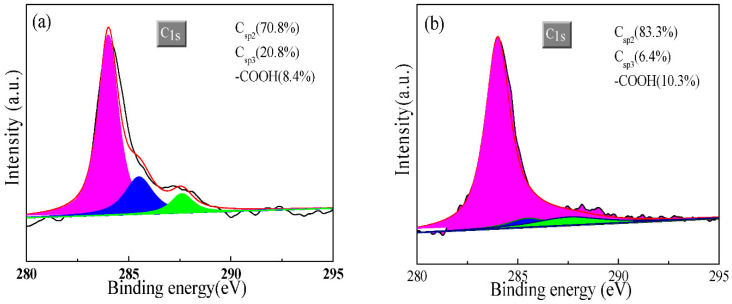
The element chemical valence of wear traces lubricated with PAO6, PAO6 + 0.1 wt% GR and PAO6 + 0.1 wt% F-GR at load 50 N and 10 mm/s for 60 min. (**a**–**c**) C1s, (**d**–**f**) O1s, (**g**) Fe2p, (**h**) N1s, (**i**) F1s.

**Table 1 nanomaterials-11-02126-t001:** Element atom contents of wear traces of disks lubricated with different oil samples at load 50 N and 10 mm/s for 60 min.

Items	Element Atom Contents (%)				
	C	O	Fe	Cr	F
PAO6	23	35.22	35.11	3.05	-
PAO6 + 0.1 wt% GR	25.67	38.16	30.13	2.18	-
PAO6 + 0.1 wt% F-GR	27.87	34.32	27.79	2.6	1.78

## Data Availability

The data used to support the findings of this study are available subject to approval from the relevant departments through the corresponding author upon request.

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
