# Peer review of "Characterization and Tribological Performances of Graphene and Fluorinated Graphene Particles in PAO"

_nanomaterials, 2021, doi:10.3390/nano11082126_

Round 1
Reviewer 1 Report
This study addresses a long standing problem concerning improving lubrication and shear performance of PAO by lubricating oil additives such as graphene or flurinated graphene nanoparticles.
The study presents a comprehensive characterization and analysis about the morphological, physical, chemical and tribological properties of these composite materials with different percentage of lubricating additives. From this analysis arises a complicated picture which can explain the large variation in results obtained with these additives. The authors provide a reasonable explanations for this behavior. Fortunately, the authors found a sweet spot in which there is a significant improvement of FG as additive. The study can pave the way for further improvement and optimization of these composite materials.
I recommend publishing this study without any corrections.
Author Response
Dear Reviewer,
Thanks for your positive comments for our manuscript.
Best regards!
Enzhu Hu
Hefei University
Reviewer 2 Report
The manuscript “Characterization and Tribological Performances of Graphene and Fluorinated Graphene Particles in PAO”, by Chen et al. reports the effects of graphene and fluorinated graphene on the lubrication characteristics of PAO6.
The manuscript provides interesting data and is well written. The characterization techniques have been chosen appropriately with the problem raised. The results are well discussed providing consistent and well-documented explanations.
However, some information regarding the textural characteristics of GR and F-GR would be useful. I think the specific surface area of the samples is a parameter that could be correlated with the other data to explain the friction and wear mechanisms.
I recommend the publication of this manuscript after minor revisions.
Author Response
Dear Reviewer,
Thanks very much for your positive comments for our manuscript. Now, we have revised our manuscript as your comments point-by-point, please see bellow,
However, some information regarding the textural characteristics of GR and F-GR would be useful. I think the specific surface area of the samples is a parameter that could be correlated with the other data to explain the friction and wear mechanisms.
Answer: We added the specific surface area of the samples in “1.2 Analysis method ”, and corresponding text was also added in “2.4 Friction and wear mechanism analysis”. Please see the red texts section.
Reviewer 3 Report
- Figure 5C must be bigger because not possible the text on the insert is not visible.
Author Response
Dear Reviewer,
Thanks for your positive comments. We have revised the Figure 5 c as your comment, please see the revised manuscript.
Best regards
Enzhu Hu
Hefei University
Reviewer 4 Report
A comprehensive study into the influence on the tribological behavior of PAO6 from the addition of graphite and fluorinated graphite was examined. The study provided promising results and in general the reviewer found the work of high quality. I would have a few comments the authors might consider in a potential revision.
effects of GR and F-GR on the lubrication characteristics of PAO6 w
(1) Abstract: Expression “A series of analysis technologies”. Please revise.
(2) Introduction, second paragraph: The sentence “Studies indicated that modified GR quickly enters the friction contact surface to produce continuous friction adsorption” would need some further explanation as it was not very clear. The phrase “friction contact surface” should be elaborated.
(3) Same paragraph: The sentence should be “Zhang [13] explored the tribological behavior”.
(4) Could the authors provide representative force-displacement curves from the tribological tests?
(5) Average wear rate seems to be smaller at higher loads (figure 5b). Please provide some explanations to the involved mechanisms.
(6) The influence of material hardness, Young’s modulus and morphology (e.g., roughness) on the coefficient of friction could be further elaborated. Fundamentally, these factors should be important in the observed results. The authors could look into the following tribological studies to further enhance their discussions:
(i) "Effect of Young’s Modulus and Surface Roughness on the Inter-Particle Friction of Granular Materials", Materials 2018, 11, 217; doi:10.3390/ma11020217.
(ii) "An experimental investigation of the microslip displacement of geological materials", Computers and Geotechnics 2019, 107, 55-67.
(7) Conclusions section could be further enhanced on some general implications from the study in tribology engineering and also with some recommendations for future research.
Author Response
Dear Reviewer,
Thanks for your positive comments, and we have revised our manuscript as point by point, the details please see bellow,
A comprehensive study into the influence on the tribological behavior of PAO6 from the addition of graphite and fluorinated graphite was examined. The study provided promising results and in general the reviewer found the work of high quality. I would have a few comments the authors might consider in a potential revision.
Effects of GR and F-GR on the lubrication characteristics of PAO6
(1) Abstract: Expression “A series of analysis technologies”. Please revise.
Answer: We revised it in Abstract and conclusions. Please see the revised manuscript.
(2) Introduction, second paragraph: The sentence “Studies indicated that modified GR quickly enters the friction contact surface to produce continuous friction adsorption” would need some further explanation as it was not very clear. The phrase “friction contact surface” should be elaborated.
Answer: We changed "the friction adsorption" into " friction adsorption film", and deleted "contact " to elaborate the " friction contact surface". The friction surface is the surface of friction pairs.
(3) Same paragraph: The sentence should be “Zhang [13] explored the tribological behavior”.
Answer: We revised it.
(4) Could the authors provide representative force-displacement curves from the tribological tests?
Answer: Thanks for reviewer's suggestion. We try our best to draw the force-displacement curves using UMT-2 tribometer. However, it was failure, because the displacement was not obtained. Besides, we focus on investigating the tribological behavior of particles in PAO6. The friction coefficient and wear rate were the important parameters for clarifing the efficiency. So, hope that our response could be understood by reviewer.
(5) Average wear rate seems to be smaller at higher loads (figure 5b). Please provide some explanations to the involved mechanisms.
Answer: You are right. We clarifiy it, please see the "2.4 Friction and wear mechanism analysis" section. In general, the high loads will cause the accumulation of friction heat on the surface of friction-pairs which will possibly promote the occurrence of tribological chemical reactions which resulted in lower wear rates for samples at high loads.
(6) The influence of material hardness, Young’s modulus and morphology (e.g., roughness) on the coefficient of friction could be further elaborated. Fundamentally, these factors should be important in the observed results. The authors could look into the following tribological studies to further enhance their discussions:
(i) "Effect of Young’s Modulus and Surface Roughness on the Inter-Particle Friction of Granular Materials", Materials 2018, 11, 217; doi:10.3390/ma11020217.
(ii) "An experimental investigation of the microslip displacement of geological materials", Computers and Geotechnics 2019, 107, 55-67.
Answer: We read the suggested papers and also added them into our manscript to clarify the friction and wear mechanism. The details please see the texts in "2.4 Friction and wear mechanism analysis" in revised manuscript.
(7) Conclusions section could be further enhanced on some general implications from the study in tribology engineering and also with some recommendations for future research.
Answer: We revised the conclusions as the reviewer 's comment. Please see the revised manuscript.